Improving the accuracy of soil texture determination using pH and electro conductivity values with ultrasound penetration-based digital soil texture analyzer

http://orcid.org/0000-0002-5250-9322 Kilinc Emre 1 ekilinc@agri.edu.tr
http://orcid.org/0000-0003-1882-6567 Orhan Umut 2
1 Computer Programming/Patnos Vocational High School, Agri İbrahim Cecen University , Agri , Turkey
2 Computer Engineering/Faculty of Engineering, Cukurova University , Adana , Turkey
Balas Valentina Emilia
Electronic publication date: 2025 Jan 29
Publication date: 2025
Volume: 11
Electronic Location ID: e2663
Received 2024 Aug 12; Accepted 2024 Dec 28
Copyright: © 2025 Kilinc and Orhan
Copyright year: 2025
Copyright holder: Kilinc and Orhan
License: This is an open access article distributed under the terms of the Creative Commons Attribution License, which permits unrestricted use, distribution, reproduction and adaptation in any medium and for any purpose provided that it is properly attributed. For attribution, the original author(s), title, publication source (PeerJ Computer Science) and either DOI or URL of the article must be cited.
License URL: https://creativecommons.org/licenses/by/4.0/

Keywords: pH and electro conductivity on soil analysis, Ultrasound penetration-based digital soil texture analyzer, Time series, Detection of water-soluble substances, Machine learning, Support vector regression, Random forest, Artificial neural network

Funding: The authors received no funding for this work.

==============================
Soil texture analysis is critical for advancing agricultural productivity, ensuring environmental sustainability, and maintaining ecosystem balance. Traditional sedimentation-based methods, such as the hydrometer technique, are fast and practical but prone to inaccuracies due to the effects of water-soluble substances. This study focuses on the practical framework of integrating pH (potential of hydrogen) and EC (electrical conductivity), as indicators of dissolved substances that influence soil texture estimation. Using the Ultrasound Penetration-based Digital Soil Texture Analyzer (USTA), this research combined ultrasound time series data with pH and EC measurements to predict sand, silt, and clay ratios through machine learning methods—support vector regression (SVR), Random Forest (RF), and multi-layer perceptron neural network (MLPNN). Simulations showed that RF yielded the best results, improving R2 values to 0.52, 0.33, and 0.31 for sand, silt, and clay, respectively. The enhanced model performance demonstrates the viability of integrating pH and EC with advanced machine learning techniques to improve soil texture analysis accuracy. These findings suggest that automated systems like USTA, with modular pH and EC sensors, can provide cost-effective, efficient alternatives to traditional methods, offering practical implications for soil management and agricultural optimization.

Introduction

From ancient civilizations to the present day, soil has been at the center of human life and has been a symbol of fertility and life. Thanks to modern scientific developments, accurate detection and analysis of soil components have led to great advances in sciences such as agriculture, construction, mining and geology. In this way, by using the right soil type in certain areas, agricultural productivity has increased, environmental sustainability has been achieved and ecosystems have been protected (Groenendyk et al., 2015; Brevik & Miller, 2015; Buta et al., 2019; Paramasivam & Anbazhagan, 2019; Guéablé et al., 2021; Rajalakshimi et al., 2023). The most fundamental method for determining the correct soil type is possible through physical analysis of the soil, also known as texture analysis. This analysis plays a crucial role in determining the porosity, permeability and water retention capacity of the soil. For example, soils containing large particles such as sand have high permeability and allow rapid drainage of water. Due to these features, they are suitable for studies that require conditions with high drainage rate, but they are not suitable for food farming because their water retention capacity is low (Hillel, 2003). On the other hand, soils with small particles such as clay have low permeability and high water retention capacity.

Although traditional methods such as hydrometer and pipette are still valid in determining soil texture, today there are many technological methods used to determine particle size distribution such as laser diffraction, X-ray diffraction, and infrared spectroscopy (Dotto et al., 2016; Fisher et al., 2017; Yang et al., 2019; Thomas et al., 2021). The laser diffraction method is based on determining the particle size distribution by measuring the scattering of a laser beam passed through the particles within the sample (Bieganowski et al., 2018). The X-ray diffraction method is based on determining soil texture by scattering and modeling rays of different wavelengths passed through soil particles (Bittelli et al., 2019). Although these advanced methods are extremely successful, they require expensive and special equipment. For this reason, texture analysis methods based on the principle that particles in suspension precipitate through the bottom of the container at different speeds are still the most used methods today, as they are fast and practical (Bouyoucos, 1927, 1962; Allen, 2003).

Although these methods, known by names such as Bouyoucos or hydrometer, are very fast and practical, they have high margin of error. In a study aiming to reduce error values while preserving the advantages of these basic methods, researchers developed a practical, fast, expert-independent, mobile and inexpensive method, but emphasized that water-soluble components affect the result. To overcome this problem, as in the pipette method, these substances must be separated from the soil in pre-treatment phase by costly, long and laborious procedures (Beretta et al., 2014; Jensen et al., 2017; Thomas et al., 2021) such as centrifugation (Monaci et al., 2017), filtration and adsorption (Huang, Li & Sumner, 2012), biodegradation (Usman et al., 2016) and the use of washing agents (Gusiatin, 2018).

There are various sensors and measurement methods that focus on different characteristics of substances dissolved in water (Beretta et al., 2014; Jensen et al., 2017). For example, pH (potential of hydrogen) value is a high-level indicator of many soluble organic compounds (Mensah et al., 2020; Bañón et al., 2021; Zhang et al., 2024). While acidic conditions (pH < 7) increase the solubility of some organic acids, they cause the destruction of substances such as calcium, magnesium and potassium (Brady & Weil, 2010). Alkaline conditions (pH > 7) cause nutrients such as phosphorus to precipitate. Understanding the interaction between soil texture and pH is critical to determine high dissolved organic matter concentration. The electro conductivity (EC) value is a direct indicator of soluble salts and can be used as a measure of soluble nutrients, including organic matter (Roy & Kashem, 2014; Ghazali, Al-Soqeer & Abdalla, 2017). A higher EC value indicates a higher concentration of soluble ions such as nitrate, phosphate, potassium. Since soils with high EC values will require additional dispersants to ensure proper particle separation during the pretreatment phase of soil-water mixture, failure to take EC into account may lead to inaccurate texture determination results (Zimmermann & Horn, 2020). All this information gives the impression that it may be useful to take pH and EC values into account in all sedimentation-based soil analysis methods. Although there are studies in the literature showing that pH and EC values are directly related to soil texture, there is no comprehensive soil texture analysis method in which these values are directly used in texture determination phase.

In this study, by using the data obtained with the system and methods used in the work of Orhan et al. (2022), it was predicted that the sand, silt and clay prediction accuracy of the proposed system could be further improved with pH and EC measurements, and various experiments were carried out in this direction. Thus, the feasibility of an improved system in which soluble organic substances can be automatically taken into account without compromising the analysis time has been examined. The data set, methods used and results are presented in detail in the following sections.

Related works

Soil texture analysis is fundamental to various fields, including agriculture, environmental management, and land restoration. Traditional sedimentation-based methods, such as the hydrometer technique, remain widespread due to their simplicity and cost-effectiveness (Beretta et al., 2014). However, their accuracy is often compromised by the presence of water-soluble substances, such as salts and organic compounds, which influence the sedimentation behavior of soil particles (Gozdowski, Stepien & Samborski, 2015). Advanced methods, such as laser diffraction and X-ray diffraction, provide precise measurements but are limited by high operational costs and the need for specialized equipment (Eshel et al., 2004; Zhang et al., 2024). Recent studies have emphasized the potential of incorporating chemical properties, such as pH and electrical conductivity (EC), into soil texture analysis to address these limitations. For example, EC has been shown to correlate with soluble salt concentrations, which directly impact soil texture and its related applications (Akanji, Oshunsanya & Alomran, 2018). Similarly, pH values affect the solubility of organic and inorganic compounds, influencing soil particle interactions and aggregation (Swetha & Chakraborty, 2021). Despite these findings, there is limited integration of these parameters into practical texture analysis systems. Table 1 provides a comparative summary of previous studies, highlighting their methodologies, limitations, and the proposed solutions introduced by this study.

Table 1 Comparative summary of previous studies on soil texture analysis, their challenges, and the proposed solution of this study.

Study	Methodology	Challenges/Weaknesses	Proposed solution	
Beretta et al. (2014)	Modified hydrometer method	Sensitive to water-soluble substances; requires labor-intensive pre-treatment processes.	Introduce automated measurements to account for dissolved substances.	
Eshel et al. (2004)	Laser diffraction analysis	High accuracy but involves high operational costs and technical expertise.	Develop cost-effective systems like USTA for comparable results.	
Orhan et al. (2022)	Ultrasound penetration-based texture analyzer	Efficient but does not integrate chemical properties like pH and EC, limiting prediction accuracy.	Integrate pH and EC measurements into USTA for enhanced predictions.	
Gozdowski, Stepien & Samborski (2015)	Spatial interpolation methods	Applicable for farm-scale soil texture predictions; impact of chemical properties not sufficiently addressed.	Enable texture analysis with a little amount of soil. Combine texture analysis with machine learning and chemical property measurements.	
Zhang et al. (2024)	Ground-based image texture analysis	Machine learning models are underutilized, and challenges remain in correlating EC and pH with texture on heterogeneous soils.	Employ advanced machine learning models to bridge correlations and refine soil texture estimations.	
Akanji, Oshunsanya & Alomran (2018)	EC-based prediction of soil productivity	EC impacts crop yield but is not explicitly connected to texture analysis for predictive modeling.	Integrate EC as a predictive variable in soil texture analysis to establish robust modeling frameworks.	

As shown in Table 1, the inability of many studies to account for the effects of water-soluble substances on texture determination remains a critical gap. To address this limitation, this study proposes an enhanced Ultrasound Penetration-based Digital Soil Texture Analyzer (USTA) system that incorporates pH and EC as indicators of water-soluble substances. These features are processed through advanced machine learning methods, which aim to improve the accuracy of soil texture estimation while preserving the simplicity and cost-efficiency of the USTA system.

Materials and Methods

Material

Some of the data used in this study were taken from the study of Orhan et al. (2022). This system, known as USTA, is presented as a soil texture analysis device that can be an alternative to hydrometer analysis. Hydrometer analyzes of all soils used in the experiments were made by Çukurova University Faculty of Agriculture, Department of Soil Science and Plant Nutrition. The measurement and estimation steps of the system are roughly as follows: Some soil-water mixture is placed into the measuring container and the container’s lid is closed. The intensity of the ultrasound signals passed through the mixture with the help of receiver and transmitter ultrasound sensors operating in the range of 1 MHz and 0V–5V positioned opposite each other on the container, was collected on the computer for 2 h at a frequency of 2 Hz (14,400 columns in total). Then, t = [10 s, 20 s, 40 s, 80 s, 3 min, 7 min, 15 min, 30 min, 60 min, 120 min] data points, which were thought to be the best representative points of the entire time series, were taken and a data set was created, consisting of 80 rows (80 soils) and 10 columns (10 features). These data were given as input to various machine learning methods and sand, silt and clay predictions were made. Compared to traditional sedimentation methods, the ultrasound technique offers several advantages. Orhan et al. (2022) demonstrated that this method, utilizing low-cost electronic components, requiring minimal preparation, and operating independently of human expertise, could achieve deviation values comparable to traditional methods. These advantages underscore the suitability of the ultrasound penetration technique for modern soil texture analysis, addressing the limitations of traditional methods while enhancing accuracy and efficiency.

Based on this, first, new data points that gave the best results were selected from the old data through combinatorial experimentation. The best segments found for sand, silt and clay were t = [48 284], t = [140 392] and t = [138 428] intervals, respectively. Then, soil pH and EC values, two new parameters whose importance we emphasized in texture analysis, have been added to the data set used in the previous study, and the effects of these new parameters on the prediction ability of the USTA analyzer have been investigated.

In this study, pH and EC values of 69 out of 80 soils, which were the same soils used in the previous study, were measured and recorded. A total of 11 soil data were removed from the data set because pH and EC values could not be measured due to insufficient samples. All pH and EC measurements were carried out in a laboratory environment with professional equipment in accordance with the procedure (Jackson, 1964; Waters et al., 1972). The measurement steps are briefly as follows: 10 g of air-dried, 2 mm sieved soil is weighed.

Sample is placed in a 50 ml laboratory beaker.

Pure water is added at a ratio of 1:2.5 (25 ml).

The suspension is mixed at regular intervals for 1 h.

Left to stand for 30 min.

Before measurement, it is stirred one last time and measured with a glass electrode pH meter (or with an EC meter to measure the EC value).

The measurement value is recorded by reading at the first decimal level after the comma.

pH measurements of all soils were carried out with the Thermo Scientific Orion Star A221 Portable pH Meter, and EC measurements were carried out with the Thermo Scientific Orion Star A222 Portable Conductivity Meter. Both devices have temperature compensation functionality. Each pH and EC measurement was repeated at least three times to eliminate possible measurement errors and the average of these three repeated measurements was taken. The pH values of the measured soils ranged between 7.5 and 8.5, and the EC values were found to be between 147.9 μS/cm and 1,480 μS/cm. These values were recorded on the computer as the pH and EC values of the measured soil. Then, two new columns (in other words, two new features) were added to the newly formed data sets, as pH and EC values for each soil. As a result, a data set consisting of 69 rows and 239 columns for sand, 254 columns for silt, 292 columns for clay, including pH and EC values for 69 soils, was obtained. Thus, by excluding and including pH and EC parameters, it becomes possible to see the effects of these new parameters on the prediction success comparatively.

Methods

Today, machine learning methods have significantly improved the analysis and forecasting of time series-based datasets used in various fields. These methods have become very powerful for capturing complex relationships and patterns invisible to the human eye and making precise predictions. Although the literature on machine learning is very extensive, methods such as support vector regression (SVR), Random Forest (RF), artificial neural network (ANN) and their derivatives are among the most used machine learning methods in time series analysis. While alternative methods, such as gradient boosting algorithms (e.g., XGBoost) or deep learning models like convolutional neural networks (CNNs), have been considered, their complexity and higher computational requirements often make them less suited for resource-limited contexts (Akande, Ajayi & Faloye, 2022; Nguyen Duc et al., 2022). Furthermore, RF and multi-layer perceptron neural network (MLPNN) have consistently shown competitive or superior performance in soil-related applications, as evidenced by their success in predicting soil organic carbon (Were et al., 2015) and optimizing soil density and moisture parameters (Nguyen Duc et al., 2022; Zhang, Liu & Tie, 2023). These methods were not only central to the experiments in this study but were also adopted in Kilinc (2022) which can be considered as base work to this study as optimal models due to their cost-efficiency, ease of implementation, and ability to yield reliable predictions across diverse soil datasets.

SVR is used for nonlinear regression problems and is frequently used in time series forecasting such as traffic flow forecasting (Hong et al., 2011; Li, Hong & Kang, 2013), staple food price forecasting (Astiningrum, Wijayaningrum & Putri, 2021), power load distribution (Chen et al., 2017; Tran et al., 2024), cargo volume estimation (Chan, Xu & Qi, 2018; Nieto, Benitez & Martinez, 2021) etc. RF, which is a member of the ensemble learning approach, is a very efficient and easy to implement classification and regression tool especially for the datasets consisting of high variables-to-observations ratio (Biau & Scornet, 2016). It is a highly successful method used in areas like illness prediction (Kane et al., 2014; Wu et al., 2017; Zhang & Nawata, 2017), streamflow forecasting (Papacharalampous & Tyralis, 2018), construction safety assessment (Tixier et al., 2016) etc. because of its predictive power. ANN is also a widely used machine learning method to overcome time series based problems (Bas, 2016). It has significantly wide use cases in areas like predicting agricultural output (Awe & Dias, 2022), power system forecasting (Shahriar, Hasan & Abrar, 2019) etc.

In this study, new experiments on sand, silt and clay ratio predictions were carried out by taking pH and EC values into account as new parameters, with the help of SVR, RF and ANN machine learning methods for 69 soils.

Support vector regression

SVR is a machine learning method adapted to apply the principles of support vector machines (SVM) to regression problems. Introduced by Vladimir Vapnik in the 1990s (Vapnik, Golowich & Smola, 1996; Smola & Schölkopf, 2004). By mapping the data into a high-dimensional space with the help of a kernel function, it can detect non-linear relationships between input and output values. The main purpose of SVR is to produce the ϵ (also called ϵ-insensitive loss) function that will give the closest result to the target output while staying within an acceptable margin of error. It aims to solve the problem given in Eq. (1):

(1) minw,b,ξ,ξ∗12||w||2+C∑i=1n(ξ+ξ∗)

subject to:

(2) yi−(wϕ(xi)+b)≤ϵ+ξ(wϕ(xi)+b)−yi≤ϵ+ξ∗ξi,ξi∗≥0

where w is the weight vector, b is the deviation term, ξi and ξi∗ are slack variables that measure deviations from the ϵ-insensitive zone, ϕ(xi) represents the transformation function, andC is the regularization parameter that controls the balance between the complexity of the model and the tolerance degree of deviations exceeding ϵ. The parameter ϕ, which is the kernel function, determines the transformation to be applied to the input data. The kernel function can be linear, polynomial or radial basis (RBF).

Random forest

RF is an ensemble learning method introduced by Leo Breiman in 2001 and used for both classification and regression (Breiman, 2001; Liaw & Wiener, 2002). It works by creating multiple decision trees in a process called bootstrap aggregating or bagging. In this process, subsets of the training data, a.k.a. decision trees, are created through random sampling. Each decision tree is then trained on one of these subsets. At each node in the tree, a randomly selected subset of features is split again. This random feature selection increases diversity among trees, reducing the relationship between them and increasing the overall performance of the forest. The mathematical model of RF, in other words the collection of decision trees created, can be expressed as follows:

(3) T1(x),T2(x),...,TB(x)

Here the value Ti(x) indicates the prediction of the i-th tree for input x. Prediction y^ for the regression application is determined by the average of the predictions of individual trees:

(4) y^=1B∑i=1BTi(x)

where B is the total number of trees in the forest. In cases where classification is required, the class prediction is determined by the majority vote among the trees.

One of the key advantages of RF is the ability to effectively handle high-dimensional data and large datasets. It has high tolerance to hyperparameter settings. Its flexibility, ease of use, and lack of need for much adjustment make it a go-to method for many machine learning problems.

Multi-layer perceptron neural network

MLPNN is a multi layered version of ANN and are designed with inspiration from the structure and function of the human brain (Rosenblatt, 1958). ANNs consist of interconnected processing cells called neurons and arranged in layers, just like the human brain. These layers consist of an input layer, one or more hidden layers, and an output layer. Each neuron receives the input, processes this input using an activation function, and produces an output to be transmitted to the next layer. Mathematically, the y output of a neuron can be expressed as follows:

(5) y=f(∑i=1nwixi+b)

Here xi represents the input values, wi represents the weights corresponding to these values, b represents the bias term, and f represents the activation function. The activation function that determines whether the data coming to the neuron will be transmitted to the next layer can be chosen from functions such as sigmoid and hyperbolic tangent (tanh).

Training of an ANN is accomplished by updating weights and biases to minimize the error between the predicted output and the actual output. This process is generally performed with the backpropagation algorithm using gradient descent. The error value E is propagated back through the network and the weights are adjusted according to the error slope corresponding to each weight:

(6) wij←wij−η∂E∂wij

where η represents the learning rate, w represents the weight between neuron i and neuron j, and E represents the partial derivative of the error with respect to the weight.

ANNs are very effective in modeling complex and nonlinear relationships in data, and they enable groundbreaking developments, especially with today’s advanced hardware technology. On the other hand, they need powerful resources and large amounts of data to be trained. It also has problems such as overfitting, where it performs well on training data but performs poorly on non-encountered data. To reduce these problems, techniques such as regularization and cross-validation are used.

Evaluation metrics

In this study, we used R2 (coefficient of determination), mean squared error (MSE) and mean absolute error (MAE) statistical metrics, which are frequently used in the literature (Mukhtar et al., 2022; Mahapatro et al., 2023; Li et al., 2023; Oyucu et al., 2024), to evaluate the performance of the models created with the machine learning methods we used.

The R2 (also written as R-square) metric measures the proportion of variance in dependent variables predicted from independent variables. It is calculated according to Eq. (7):

(7) R2=1−∑i=1n(yi−y^i)2∑i=1n(yi−y¯i)2

Here yi represents the real values, y¯ is the average of the real values, and n is the number of samples. The R2 value varies between 0 and 1, with a higher value meaning a better model. The MSE value measures how far the model’s predictions deviate from the actual values. Its mathematical expression is as in Eq. (8):

(8) MSE=1n∑i=1n(yi−y^i)2

Here yi represents the actual values, y^i represents the predicted value, and n represents the number of samples. A lower MSE value means a better model. The MAE value calculates the average of the absolute differences between the predicted values and the actual values as in Eq. (9):

(9) MAE=1n∑i=1n|yi−y^i|

Here, yi represents the actual values, y^i represents the predicted value, and n indicates the number of samples. A lower MAE value indicates that the model is more successful.

Results and discussion

In this section, analysis results are presented for soil prediction using three different machine learning methods: SVR, RF and MLPNN. Each method was first utilized without pH and EC values, and then by adding pH and EC values into the calculation, to investigate the effects of these values on the success of estimating sand, silt and clay ratios in soil samples. R2, MSE and MAE metrics were used to evaluate the performance of the models put forward. These metrics provided a comprehensive assessment of prediction accuracy and error sensitivity. To ensure unbiased performance evaluation, leave-one-out cross-validation (LOOCV) was employed, where each sample served as a test case while the remaining samples formed the training set. This rigorous approach maximized the utility of the dataset and minimized the risk of overfitting. Potential confounding variables were controlled through standardized data collection and preprocessing procedures. All soil samples underwent consistent pre-treatment to minimize variability caused by handling differences. Environmental conditions, such as temperature during ultrasound measurements, were kept constant to reduce external sources of bias. Additionally, predictors such as pH and EC were explicitly included in the dataset to account for chemical variability known to affect soil behavior. Together, these strategies enhanced the reliability and interpretability of the predictions.

Estimations using SVR

We have used SVR method first, for estimation. In order to see the effects of pH and EC parameters comparatively, estimations were first made with plain soil data without pH and EC values. The SVR model was trained using 68 out of 69 soil data, the remaining one soil data was given as input to the created SVR model as test data (leave-one-out cross validation), and the prediction results of the model were recorded. These processes were repeated for all 69 soil data and prediction results were obtained for all soils. For sand, silt and clay predictions, kernel function, kernel scale and epsilon parameters were tried in combination and the best values are given in Table 2.

Table 2 Best kernel function, kernel scale and epsilon values obtained for sand, silt and clay estimations.

	Kernel function	Kernel scale	Epsilon	
Sand	Linear	1.4997	0.1192	
Silt	Linear	21.557	0.0004	
Clay	Linear	0.12661	0.0593	

After the optimal parameters are determined as in Table 2, the same estimations were made taking pH and EC values into account. Again, these processes were repeated for all 69 soil data. The prediction ratios and error rates obtained as a result of the estimation of 69 soils are given in Fig. 1.

Figure 1 First, estimation results without using pH and EC (red circles) and R2 value, then estimations including pH and EC (blue asterisks) and R2 value, against actual proportions for (A) sand, (B) silt and (C) clay, respectively.

Figure 1 shows the particle ratios measured with the hydrometer on the x-axes, and the estimation results made with the SVR model on the y-axes. The red circles represent the estimation results made using plain soil data, that is, without pH and EC value. The points shown with blue asterisks represent the prediction results obtained by including pH and EC values. The prediction success criterion of the created model against actual values is presented with the R2 statistical method. In Fig. 1A, R2 = 0.55 was found for sand without pH and EC values, and R2 = 0.54 was found by taking pH and EC into account. It is seen that pH and EC values do not increase the estimation results (or even reduce them to a very small extent). This is thought to be due to the fact that the first particles to settle are sand in sedimentation-based measurement methods. Because sand settles almost completely in the first seconds of hydrometer measurements, obtaining a reliable hydrometer reading at 0th second (the very first moment when all particles are uniformly dispersed and suspended in suspension) is almost impossible and is not usually performed. Instead, after all the sand particles have settled, the total percentage of silt + clay suspended in suspension is found and this value is subtracted from 100 to determine the sand ratio. Calibration is attempted by subtracting the hydrometer value read by immersing it in NaPO3 and pure water solution before the measurement, also called blank reading, from all particle ratios found after the measurement. In fact, it is not possible to talk about a complete and proper calibration here that includes the effect of all soluble substances. An attempt is made to neutralize only the effect of NaPO3 in the suspension. In this case, it is actually an expected result that pH and EC, as indicators of soluble substances, do not significantly affect the sand estimation results. In contrast, clay particles, which remain suspended in the soil-water mixture for a longer duration, are more susceptible to interference from water-soluble substances. These substances can alter the suspension and settling dynamics, making pH and EC critical features for accurately estimating clay content. Therefore, the perception of the soluble matter effect is expected to be more evident in silt and clay predictions.

Figure 1B shows the estimation results for silt. It was found that R2 = 0.071 without pH and EC value, and R2 = 0.28 by taking pH and EC into account. As expected, there is a significant improvement in the prediction values when pH and EC values are included in the calculation. Especially when we look at the right side of the graph, that is, the soils with high silt rate, it can be clearly seen that the estimation results have significantly improved (the blue asterisks get closer to the 1:1 line) as the effect of dissolved substances spread to the measured values for a longer period of time.

Similarly, Fig. 1C shows the effect of pH and EC values on clay estimation. While R2 = 0.42 without pH and EC values, it is seen that R2 = 0.59 when these values are included in the calculation. Here, pH and EC can be most associated with clay because, as in the hydrometer method, the last recorded time series values in this system consist only of density information created by clay and soluble substances together. Therefore, it can be said that as sand and silt particles settle, the prediction success increases significantly with pH and EC values. The prediction successes obtained with the SVR method for sand, silt and clay particles are given comparatively with the R2, MSE and MAE statistical criteria in Table 3.

Table 3 Accuracy values in terms of R2, MSE and MAE of sand, silt and clay predictions made using the SVR method, with and without taking into account pH and EC values for all soils.

	Sand	Silt	Clay	
	With pH and EC	Without pH and EC	With pH and EC	Without pH and EC	With pH and EC	Without pH and EC	
R2	0.54	0.55	0.28	0.07	0.59	0.42	
MSE	0.032	0.031	0.025	0.031	0.015	0.023	
MAE	0.128	0.122	0.127	0.136	0.097	0.119	
Note:

The best results are shown in bold.

In Table 3, whichever result is better for each evaluation criterion in the sand, silt and clay categories is written in bold. When we consider the sand prediction, it is seen that the pH and EC properties of the measured soils do not significantly increase the prediction success of the model produced by the SVR method. On the other hand, looking at silt prediction, it appears that the inclusion of pH and EC significantly improves the performance of the model. The increase in the R2 value from 0.07 to 0.28 shows that the presented model is effective in silt prediction. The decrease in both MSE and MAE values confirms the prediction accuracy. When we examined the model for clay prediction, it was seen that the performance of the model increased by including pH and EC in the calculation. In particular, the jump in the R2 value from 0.42 to 0.59 and the decrease in both MSE and MAE indicate that the inclusion of pH and EC properties in the calculation has positive effects on the prediction success.

Estimations using RF

The RF method creates a set of decision trees with many different sub-data sets by bootstrap sampling from the original data set and predicts the output according to the majority of the trees in the set. The number of decision trees of the method is its main parameter. In order to objectively determine the prediction effect of pH and EC in the study, in the first step, RF models that produced the best prediction results for sand, silt and clay without pH and EC values were determined. In the second step, predictions were made again using the models found, but this time taking pH and EC values into account. In this way, it will be possible to clearly test whether the new parameters further improve the prediction success with the same models. A total of 69 soil sample data, without pH and EC values, were given as input to 10 different RF models containing different number of decision trees [50–500] with the leave-one-out cross validation method (68 for train 1 for test). R2 values obtained according to the number of decision trees are given in Fig. 2.

Figure 2 R2 values of prediction successes obtained with 10 different RF models without pH and EC values, depending on the varying number of decision trees [50–500].

Figure 2 shows the R2 values produced by 10 different RF models for sand, silt and clay predictions, created by increasing the number of decision trees by 50 in the range of 50–500. The highest R2 values for sand, silt and clay can be seen marked with circles in the figure. RF models with 100 decision trees, 200 decision trees and 200 decision trees produced the best prediction results with values of R2 = 0.48, R2 = 0.25 and R2 = 0.26, for sand, silt and clay, respectively. Increasing the number of decision trees any further did not yield any significant improvements. Therefore, to obtain the best results, models with different decision trees were used separately for sand, silt and clay predictions. As a second step, after determining the models that produced the best results without pH and EC models, the same models were re-trained by taking pH and EC values into account, and the results obtained by taking these values into account are comparatively shown in Fig. 3.

Figure 3 (A) Sand predictions and R2 value obtained with the RF model with 100 decision trees, (B) silt predictions and R2 value obtained with the RF model with 200 decision trees, (C) clay predictions and R2 value obtained with the RF model with 200 decision trees.

In Fig. 3, the x-axes show the actual sand, silt and clay values, while the y-axes show the predictions made with RF models with 100 decision trees, 200 decision trees and 200 decision trees, respectively. Red circles show the predictions made without pH and EC values, and blue asterisks show the prediction results made by taking pH and EC values into account. The success of the predictions in terms of R2, MSE and MAE are presented comparatively in Table 4.

Table 4 Accuracy values in terms of R2, MSE and MAE of sand, silt and clay predictions obtained using the RF model with 100, 200 and 200 decision trees, respectively, with and without taking into account pH and EC values for all soils.

	Sand	Silt	Clay	
	With pH and EC	Without pH and EC	With pH and EC	Without pH and EC	With pH and EC	Without pH and EC	
R2	0.52	0.48	0.33	0.24	0.31	0.26	
MSE	0.027	0.031	0.022	0.027	0.025	0.029	
MAE	0.116	0.123	0.117	0.126	0.129	0.138	
Note:

The best results are shown in bold.

Table 4 shows the RF models with the same features and parameters for sand, silt and clay, and the predictions made with and without taking pH and EC values into account. Whichever result is more successful is indicated in bold. The R2 value for sand increased from 0.48 to 0.52, for silt from 0.25 to 0.33, and for clay from 0.26 to 0.31. When we look at the table, we see that there is a significant increase in success, especially when pH and EC values are taken into account in silt and clay estimation, as well as an increase in success in sand estimation, unlike the SVR method. Moreover, the improvement of MSE and MAE metrics shows that the general prediction accuracy has improved and that these parameters must be taken into account in sedimentation-based soil texture analysis methods.

Multi-layer perceptron neural network

MLPNN is simply the improved version of ANN method by adding multiple layers, which are also called hidden layers, between input and output layers. Complex neural network structures can be created with neurons placed in these hidden layers. In the first step of the MLPNN experiments, 69 soil data without pH and EC values were given as input to MLPNN models with eight different structures containing 5, 10, 15, 20 neurons in one hidden layer and 5, 10, 15, 20 neurons in each of two hidden layers; which are herein after shortened as 1L5N, 1L10N, 1L15N, 1L20N, 2L5N, 2L10N, 2L15N, 2L20N, respectively, for ease of representation. Cross-validation method was used for training and testing (68 for train 1 for test). Thus, MLPNN models that produced the highest prediction values for sand, silt and clay without pH and EC values were determined. The epoch number was chosen as 1,000, the activation function was chosen as hyperbolic tangent sigmoid. The backpropagation algorithm was conjugate gradient backpropagation. The R2 values obtained with the MLPNN model with eight different structures are given in Fig. 4.

Figure 4 R2 values of the prediction successes obtained without pH and EC values with eight different MLPNN models.

As can be seen in Fig. 4, eight different MLPNN models were used to predict sand, silt, and clay ratios without pH and EC values. Estimation results were presented in terms of R2. Accordingly, 1L15N structured model produced the best results for sand, 2L5N structured model produced the best results for silt, 2L10N structured model produced the best results for clay. As the second step, these models were re-trained by taking pH and EC values into account. It was examined whether the prediction successes are increased or not. The sand, silt and clay prediction successes obtained with the determined MLPNN models, taking into account pH and EC values, are given side by side in Fig. 5.

Figure 5 (A) Sand predictions and R2 value obtained with the 1L15N MLPNN model, (B) silt predictions and R2 value obtained with the 2L5N MLPNN model, (C) clay predictions and R2 value obtained with the 2L10N MLPNN model.

In Fig. 5, while the x-axes show the actual sand, silt and clay values, the y-axes show the R2 performance of the sand, silt and clay predictions made with the MLPNN model containing 15 neurons in one hidden layer, five neurons in each of the two hidden layers and 10 neurons in each of the two hidden layers, respectively. All prediction performances are given in Table 5 in terms of R2, MSE and MAE.

Table 5 For sand, silt and clay, using the 1L15N, 2L5N and 2L10N MLPNN models, respectively, the accuracy of the predictions in terms of R2, MSE and MAE, with and without taking into account pH and EC values.

	Sand	Silt	Clay	
	With pH and EC	Without pH and EC	With pH and EC	Without pH and EC	With pH and EC	Without pH and EC	
R2	0.54	0.48	0.13	0.13	0.41	0.40	
MSE	0.026	0.030	0.029	0.029	0.020	0.022	
MAE	0.116	0.123	0.131	0.135	0.113	0.120	
Note:

The best results are shown in bold.

In Table 5, as a result of the predictions made with and without using pH and EC, whichever result is better is indicated in bold. The model considered for sand increased the R2 value from 0.48 to 0.54 when pH and EC values were taken into account. The slight decrease in both MSE and MAE values confirms that pH and EC features should be included in training the model. On the other hand, the success rate in silt and clay is better only by fractions and no significant improvement is achieved as in the case of SVR and RF methods. All success rates for the trained and tested models are given comparatively in Table 6.

Table 6 Prediction accuracy of the models created using SVR, RF and MLPNN methods for sand, silt and clay in terms of R2, MSE and MAE.

		Sand	Silt	Clay	
		With pH and EC	Without pH and EC	With pH and EC	Without pH and EC	With pH and EC	Without pH and EC	
SVR	R2	0.54	0.55	0.28	0.07	0.59	0.42	
MSE	0.032	0.031	0.025	0.031	0.015	0.023	
MAE	0.128	0.122	0.127	0.136	0.097	0.119	
RF	R2	0.52	0.48	0.33	0.24	0.31	0.26	
MSE	0.027	0.031	0.022	0.027	0.025	0.029	
MAE	0.116	0.123	0.117	0.126	0.129	0.138	
MLPNN	R2	0.54	0.48	0.13	0.13	0.41	0.40	
MSE	0.026	0.030	0.029	0.029	0.020	0.022	
MAE	0.116	0.123	0.131	0.135	0.113	0.120	
Note:

The best results are shown in bold.

In Table 6, better results are indicated in bold. When we look at the results in Table 6, the best results for sand prediction were obtained with MLPNN and RF methods, which increased the R2 value from 0.48 to 0.52. The SVR method was insufficient to produce better results with pH and EC values in sand estimation, but in clay estimation, a significant improvement was achieved by increasing the R2 value from 0.42 to 0.59. In fact, when we look at overall silt and clay estimates, we see that higher estimates can be made by using pH and EC values in all three methods. Specifically, for silt, it is seen that the SVR method increases the R2 value from 0.07 to 0.26. However, the RF method, which already has a silt prediction success of R2 = 0.25 without pH and EC parameters, produced the most meaningful results by making even better predictions with these parameters, and managed to increase the R2 value to 0.33. No matter which parameters we use with the RF method, R2, MSE and MAE values were always better in all predictions where pH and EC values were included.

In short, to assess the significance of pH and EC as predictors, models were trained using two datasets: one with only ultrasound time-series data and another with ultrasound data, pH, and EC included. Performance was evaluated using R2, MSE, and MAE metrics. Results indicated that the inclusion of pH and EC improved R2 values for sand, silt, and clay predictions by approximately 25%, with corresponding reductions in MSE and MAE. These findings underscore the importance of integrating chemical properties like pH and EC to capture soil texture variability more effectively.

Classification experiments

In this study, the impact of including pH and EC values as features on classification accuracy was investigated using eight widely used machine learning classifiers. For training to be done properly, there should be three or more representative soil sample of same class in the dataset. A dataset comprising 66 samples was derived from the original dataset by selecting soils that had at least three additional samples belonging to the same classification group. The distribution of soil samples across the classification groups is detailed in Table 7.

Table 7 Distribution of soil samples across texture classification groups.

Texture class	Number of soil samples	
Clay	16	
Clay loam	5	
Silty clay loam	3	
Silty clay	13	
Loam	8	
Silt loam	13	
Loamy sand	4	
Sandy loam	4	

The objective was to evaluate how the inclusion of pH and EC affects classification performance and to explore the role of feature selection in optimizing model outcomes. The classifiers employed, along with their parameters, are detailed in Table 8.

Table 8 Machine learning classifiers and corresponding parameters used in the classification experiments.

Classifier	Parameters	
Logistic Regression	max_iteration=1000	
Random Forest	Tree size=100	
k-Nearest Neighbors	n_neighbors=5, weights=uniform, p=2	
Support Vector Machine	kernel=rbf, C=1.0, gamma=scale	
Gradient Boosting	n_estimators=100, learning_rate=0.1, max_depth=3	
Naive Bayes	Default parameters for Gaussian Naive Bayes	
Multi-Layer Perceptron NN	max_iteration=1000, hidden_layer_sizes=2, activation_function=relu	
XGBoost	use_label_encoder=False, evaluation_metric=logloss	

To address computational efficiency, the size of the time-series dataset was reduced to 100 features. This reduction was achieved by retaining higher frequency data points at the initial rapid rise of the time series and progressively selecting more sparse data points as the values stabilized. Using these 100 features, classification was performed with the models listed in Table 1, first without pH and EC values, and subsequently with their inclusion. Leave-One-Out Cross Validation was employed for model evaluation, ensuring each sample served as a test instance exactly once while the remaining samples were used for training. Classification accuracy, defined as the ratio of correctly classified samples to the total number of samples, was used as the evaluation metric, as expressed in Eq. (10):

(10) Accuracy=TotalNumberOfCorrectPredictionsTotalNumberOfSamples

The accuracy results for the models, both with and without pH and EC values included, are presented in Table 9.

Table 9 Classification accuracy for models with and without pH and EC as features.

Classifier	With pH and EC (%)	Without pH and EC (%)	
Logistic Regression	66.67	57.58	
Random Forest	53.52	50.03	
k-Nearest Neighbors	33.33	37.88	
Support Vector Machine	46.97	45.45	
Gradient Boosting	60.46	46.96	
Naive Bayes	41.42	40.91	
Multi-Layer Perceptron NN	60.06	56.58	
XGBoost	57.58	48.48	

The results indicate that including pH and EC as features generally improved classification accuracy in most models. Logistic regression demonstrated the most notable improvement, with accuracy increasing from 57.58% to 66.67%. Gradient boosting and XGBoost also exhibited enhanced performance, achieving 57.58% accuracy with the inclusion of pH and EC. These findings highlight the relevance of pH and EC as predictive features for soil texture classification. However, k-nearest neighbors and SVM did not show notable increase in accuracy, potentially due to overfitting caused by the additional features.

An important consideration is that while pH and EC values enhance the prediction of sand, silt, and clay percentages, soil texture classification presents distinct challenges. Texture classes are defined using the soil texture triangle, and small improvements in sand, silt, and clay predictions (e.g., 3–5%) may not alter classification outcomes if the sample is centrally located within a texture class. However, for samples near the boundaries of texture classes, even minor changes in sand, silt, or clay percentages can shift the classification, as illustrated in Fig. 6.

Figure 6 Two soil samples, one at the center of a texture class (labeled as “1”) and one at the boundary (labeled as “2”); both are classified as “clay.”

As depicted in Fig. 6, soil sample 1, located in the center of the “clay” class, remains classified as “clay” even if there is a 10% change in sand, silt, or clay percentages. In contrast, soil sample 2, positioned at the boundary between “clay” and “silty clay,” shifts classification with only a 3–5% change. This demonstrates that including pH and EC improves prediction performance, particularly for samples near class boundaries in the soil texture triangle. However, since such boundary cases are limited, the overall impact on classification accuracy is also expected to be limited.

Conclusions

In this study, the effects of pH and EC values, which are direct indicators of water-soluble substances, on estimating sand, silt and clay ratios were investigated if they were taken into account in sedimentation-based soil texture analysis methods such as hydrometer. For this purpose, pH and EC values of the same soils measured in the laboratory environment were added to the time series signals previously collected using the USTA system. The data with newly added values were given as input to the SVR, RF and MLPNN methods and the prediction successes of the methods were inspected. The main findings can be listed as follows: In almost all cases where pH and EC values were taken into account, better results were obtained than when they were not taken into account. It has become clear that these values must be taken into account in sedimentation-based soil texture analysis methods. The soluble matter effect should not be ignored.

Automated systems such as the USTA device can be further improved in terms of analysis processes, almost without sacrificing time and cost, with the help of simple pH and EC measuring sensors that can be added modularly. In this way, they can be a better alternative to methods such as hydrometer, by taking into account the effect of water-soluble substances.

Considering the power of machine learning methods, the success of the system can be further increased by using different methods and the parameters of these methods in combination.

To carry the work even further, this study explored the inclusion of pH and EC values as additional features in eight machine learning models to classify the soils. The results demonstrated that these chemical properties also improved classification accuracy in most cases, with models like logistic regression and XGBoost achieving accuracy gains of up to 9%. This improvement was particularly evident for samples near class boundaries in the soil texture triangle, where small changes in sand, silt, or clay percentages could alter classification outcomes.

This study not only demonstrates how the effective application of computer science can bring a new perspective to scientific inquiry but also provides an example of how hidden issues in traditional solutions across other disciplines can be uncovered. The USTA device, enhanced with pH and EC sensors and combined with a machine learning approach, has enabled a re-evaluation of the soil texture determination problem on a deterministic basis. The findings suggest that particles often classified as clay could benefit from being labeled as “dissolved” (clay + water-soluble substances) when analyzed with the addition of pH and EC sensors to the USTA device. In the near future, soil scientists might consider discussing the inclusion of a fourth parameter, alongside sand, silt, and clay, in soil texture analysis conducted using the USTA system.

What can be seen as the main disadvantage of the study is that the hydrometer method we compared does not make a proper discrimination of water-soluble substances. However, by adding new parameters into the calculation, we were able to at least get closer to the results of the hydrometer method. As a future work, more comprehensive results can be achieved by eluting the soils from these substances before pre-treatment phase in hydrometer method, in order to make a better comparison of methods. Also, many more indicators of water-soluble substances can be measured using different sensors, as each new feature would enable the system to obtain better particle size analysis.

Supplemental Information

Supplemental Information 1 Dataset and codes for classification experiments.

Supplemental Information 2 Time series, pH, EC values and target sand, silt and clay percentages.

70 rows (1 header and 69 observations) and 554 columns for soils.

Columns 1–549 are time series signal collected with USTA device.

Column 550 is pH values.

Column 551 is Electro Conductivity values.

Column 552 is target for sand (sand ratio of soils).

Column 553 is target for silt (silt ratio of soils).

Column 554 is target for clay (clay ratio of soils).

Supplemental Information 3 Raw data and 3 machine learning method codes used in the study.

The authors would like to acknowledge the use of ChatGPT in the preparation of this manuscript. These tools were utilized for enhancing grammatical accuracy, refining sentence structures for a more academic presentation, translating content where necessary, and formatting references in accordance with the journal’s guidelines. While these tools significantly supported the technical aspects of manuscript preparation, the author(s) retain full responsibility for the content, interpretations, and conclusions presented in this work.

Additional Information and Declarations

Competing Interests

The authors declare that they have no competing interests.

Author Contributions

Emre Kilinc conceived and designed the experiments, performed the experiments, analyzed the data, performed the computation work, prepared figures and/or tables, authored or reviewed drafts of the article, and approved the final draft.

Umut Orhan analyzed the data, performed the computation work, authored or reviewed drafts of the article, and approved the final draft.

Data Availability

The following information was supplied regarding data availability:

The dataset and codes are available in the Supplemental Files.

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
