# Peer review of "Improving the accuracy of soil texture determination using pH and electro conductivity values with ultrasound penetration-based digital soil texture analyzer"

_PeerJ Computer Science, doi:10.7717/peerj-cs.2663_

## Round 0.1 · original submission · Major Revisions

The paper must be improved. Please address all comments in detail

·

Basic reporting

Please read the comments in section 4

Experimental design

Please read the comments in section 4

Validity of the findings

Please read the comments in section 4

Additional comments

Authors must avoid writing extra contents in the abstract. I would recommend to re-draft it according to the following guidelines.

The first two lines must encompass the context of the study and the research problem, further two lines must be covered the objective of the papers with unfolding the description of the title. In the next 2 to 4 lines the methodology will be covered. Afterward, the next two lines are for result and performance. In these lines, the author must define how the results and performance are being achieved, for instance, by conducting either simulation or physical implementation. Please mention the name of the simulation or the physical method. The result statistics must be mentioned in the last two lines and either in percentage or with real-time values.

The "MATERIAL AND METHODS" section needs the massive improvement. This section must be separated from literature review and a seperate section with "Literature Review" should be added. Further, this section should be written according to the following recommendations.

This section must have discussion of any shortcomings or challenges in prior research. If the previous work is free of challenges and there is no problem, what motivated the authors to conduct this study? According to the guidelines, authors should review the literature review. They should also point out the problems in the previous study, compare the proposed solution, and explain how the proposed solution is best applicable. Additionally, please include a table summarizing related work, weaknesses, and the proposed solution.

Further concerns have given as

How do authors justify the choice of machine learning methods (SVR, RF, MLPNN) over other potential algorithms, and what specific criteria did use to evaluate their performance in the context of soil texture analysis?

Can authors elaborate on the statistical methods used to validate the robustness of findings, particularly in relation to the control of confounding variables that may affect the accuracy of soil texture predictions?

Authors mentioned the addition of pH and EC values as new features. How did this determine the significance of features in improving prediction accuracy, and what metrics did use to quantify this improvement?


how the ultrasound penetration technique works in the context of soil texture analysis, and what specific advantages it offers over traditional sedimentation methods?

While anayzing references I found reference [1] which is not aligned with the objective of this study.

·

Basic reporting

Please improve your Abstract. Please use the following checklist to ensure you have included the necessary information: (1) Background and research question(s): 1-2 sentences; (2) Theoretical or conceptual framework; (3) Research Design and Methodology; (4) Results/Key findings: 3-5 sentences. (5) Implications of your study (1-2 sentences).


Check some grammatical and spelling errors.

This manuscript's choice of reference selection needs to be updated as some references are too old. Taking into consideration the recent development of various research studies, For example, Bowman and Hutka, 2002; Jones and Parsons, 1972; (Rauch and Drewes, 2004;

The first paragraph of the introduction (Lines 33 - 39) does show a proper introduction to the study. Hence, I suggest modification and detailed information about what necessitates the study "Motivation."

Also, there are inconsistencies in the paragraph lines throughout the manuscript. Kindly check and ensure consistency. Also consider merging the short paragraphs to make it coherent in reading

Experimental design

Highlight the research gap in previous studies and clearly state the novelty of this research.

Fig. 1 should be transferred to the "Result and Discussion" section, which will be more relevant.

Validity of the findings

In the conclusion section, mention the implications of your research and its contributions to scientific knowledge. Relate your work to global interests and its worldwide importance to engage readers.

Additional comments

- The paper is well-written, and the conclusions are worthy of publication. However, some areas need further clarification before publishing. Therefore, I recommend a minor revision and resubmission.

-

Reviewer 3 ·

Basic reporting

I carefully reviewed the article submitted to PeerJ by Kilinc and Orhan (2024) titled “Improving the accuracy of soil texture determination using pH and Electro Conductivity values with ultrasound penetration-based digital soil texture analyzer”.
The effect on the estimation of clay, silt and sand was examined by adding easy-to-determine pH and EC analyzes in the soil to the device designed as a soil texture determination technique previously performed by Orhan, et al. (2022) and the technique called USTA (Ultrasound Penetration-Based Digital Soil Texture Analyzer).

Experimental design

In soil science, modeling or pedotransfer functions are carried out to make easier and cheaper analyzes and to predict difficult ones, instead of costly and costly analyzes.
In this study, the design of the device for targeted soil texture analysis is a technically important advance. And some such devices are now safe to use on the market. An example: https://metergroup.com/products/pario/

Validity of the findings

Given the parsimony of models in the framework of soil science and in the modeling framework of computer science, it may seem logical to include pH and EC analyses for an analysis such as soil texture.
However, when Figure 2 was carefully examined, the addition of pH and EC for the sand percentage did not provide any added value in the model accuracy criteria, especially with the SVR model. Very similar phenomena exist in other RF and MLPN.
And the most important feature, as the authors emphasize in their article, is that the soil texture is determined as % clay, silt sand in laboratory, but they are aggregated in "12 texture classes" as texture classes. https://www.nrcs.usda.gov/resources/education-and-teaching-materials/soil-texture-calculator

Additional comments

In this direction, a methodology specific to texture classes should be developed in terms of estimation on classes and the results should be presented. I must suggest that the paper be rejected in its current form, but it can be re-evaluated after methodological advances have been made.
Best regards,

---

## Round 0.2 · accepted · Accept

The paper was improved very well and can be accepted.

·

Basic reporting

Please read the comments in "Additional comments" section

Experimental design

Please read the comments in "Additional comments" section

Validity of the findings

Please read the comments in "Additional comments" section

Additional comments

All concerns have been addressed by the authors.

·

Basic reporting

Authors has made some modifications as suggested before

Experimental design

Done ae expected

Validity of the findings

In line with the previous comments

Reviewer 3 ·

Basic reporting

I carefully reviewed the revised version of article submitted to PeerJ by Kilinc and Orhan (2024) titled “Improving the accuracy of soil texture determination using pH and Electro Conductivity values with ultrasound penetration-based digital soil texture analyzer”.

Experimental design

The authors provided feedback on all issues reported.

They also provided feedback where relevant.

They also conducted a classification-based procedure, in particular, and provide important insights.

Validity of the findings

Now both regression and classification based modeling approaches have been applied. In its current form, the literature may be relevant in due time.

Additional comments

-